# Multiple Effects of High Surface Area Hollow Nanospheres Assembled by Nickel Cobaltate Nanosheets on Soluble Lithium Polysulfides

**DOI:** 10.3390/molecules28041539

**Published:** 2023-02-05

**Authors:** Jun Pu, Xiaomei Zhu, Jie Wang, Shaomeng Yu

**Affiliations:** 1Key Laboratory of Functional Molecular Solids, Ministry of Education, College of Chemistry and Materials Science, Anhui Normal University, Wuhu 241002, China; 2Anhui Provincial Engineering Laboratory for New-Energy Vehicle Battery Energy-Storage Materials, Anhui Normal University, Wuhu 241002, China; 3College of Chemistry and Chemical Engineering, Anqing Normal University, Anqing 246000, China

**Keywords:** hollow nanospheres, separator modification, nickel cobaltate, lithium polysulfides

## Abstract

Inhibiting the shuttle effect of soluble polysulfides and improving slow reaction kinetics are key factors for the future development of Li–S batteries. Herein, edelweiss shaped NiCo_2_O_4_ hollow nanospheres with a high surface area were prepared by a simple template method to modify the separator to realize multiple physical constraints and strong chemical anchoring on the polysulfides. On one hand, the good electrolyte wettability of NiCo_2_O_4_ promoted the migration of Li-ions and greatly improved the dynamics. On the other hand, mesoporous NiCo_2_O_4_ nanomaterials provided many strong chemical binding sites for loading sulfur species. The hollow structure also provided a physical barrier to mitigate the sulfur species diffusion. Therefore, the modified separator realized multiple physical constraints and strong chemical anchoring on sulfur species. As a result, the sulfur cathode based on this composite separator showed significantly enhanced electrochemical performance. Even at 4 C, a high capacity of 505 mAh g^−1^ was obtained, and about 80.6% could be retained after 300 cycles.

## 1. Introduction

The lithium–sulfur (Li–S) battery, as a new electrochemical system, has been widely concerned in the market of secondary energy storage. It not only has a rich source of raw materials, but also has ultrahigh theoretical specific capacity (1675 mAh g^−1^) and energy density (2600 Wh kg^−1^), which is several times more than the conventional Li-ion battery with LiCoO_2_ or LiFePO_4_ as the cathode [1,2]. Unfortunately, the large-scale commercialization of Li–S batteries still faces several challenges such as the poor conductivity of sulfur, dissolution of lithium polysulfides (LiPSs), and slow reaction kinetics [3,4,5,6]. Among them, the shuttle effect caused by soluble LiPSs (Li_2_S_n_, 4 ≤ n ≤ 8) can cause an irreversible loss of active material and a rapid decline in capacity [7]. How to solve the shuttle effect is the key to its future development [8].

In principle, the shuttle effect is the diffusion of LiPSs back and forth between the positive and negative electrodes. The separator of the cell is the only channel through which these behaviors of LiPSs occur. If the separator is designed so that it does not allow LiPSs to pass through, the shuttle effect will be fundamentally resolved. Obviously, the commercial polypropylene (PP) separator is not up to this difficult task due to the large-pore framework [9].

In recent years, many studies have shown that the initial PP separator can be modified (separator engineering) to obtain different inhibition abilities of LiPS diffusion [9,10,11]. Carbon based materials (carbon nanofibers, mesporous carbon, etc.) were first used to modify the Li–S separator [12,13]. However, the “non-polar” carbon has a limited effect on the shuttle of LiPSs, because its interaction with the latter is in the category of van der Waals forces [14,15]. The carbon coating with a high specific area on separator is only a barrier layer in the physical sense. The soluble LiPSs will still be shed from the carbon substance during long or high load electrochemical cycles. In addition, carbon materials cannot effectively solve the problem of slow LiPS conversion kinetics. Therefore, transition metal oxides with a “polar surface” are increasingly being studied. It has a good affinity with LiPSs and can effectively anchor the latter. More importantly, some metal oxides have a certain catalytic effect, which can promote the LiPS reaction. Different oxides including MnO_2_, Fe_2_O_3_, and NiO are used in Li–S batteries [16,17,18]. For instance, Wang et al. coated a functional MnO_2_/graphene oxide/carbon nanotube composite interlayer on a PP separator in Li–S batteries. Combined with the anchoring effect of MnO_2_ on LiPSs and the stability of carbon materials, the modified electrode can effectively inhibit the diffusion of LiPSs [16]. Among these oxides, the ternary metal oxide composed of two different metal cations shows higher electronic conductivity and electrochemical activity than mono-metal oxides due to its complex chemical composition and the synergistic action [19,20]. Unfortunately, the low surface area and heavy block are not conducive to the energy density. It can be seen that the correct design of transition metal oxide as a modified material for a Li–S separator still faces many challenges and is worth exploring.

Herein, the edelweiss shaped nickel cobaltate (NiCo_2_O_4_) hollow nanospheres were prepared by a simple template method (Figure 1a). The structure consisted of lots of ultra-thin NiCo_2_O_4_ nanosheets, which showed a specific surface area of up to 281 m^2^ g^−1^, meaning that it has a more active area as a separator modification material. The high surface area adsorbed more sulfur species and closed them in the inner cavity of NiCo_2_O_4_, further inhibiting the migration of soluble LiPSs. In addition, the rich Ni–Co sites provided a good catalytic path for the rapid LiPS conversion. Unlike the conventional PP separator cell (Figure 1b), the battery based on such a NiCo_2_O_4_ coated separator (defined as NiCo_2_O_4_@PP) obtained an excellent electrochemical performance (Figure 1c). A high reversibility of 1386 mAh g^−1^ could be obtained at 0.2 C. When the current density was increased to 4 C, the capacity was still as high as 505 mAh g^−1^ and remained about 80.6% after 300 cycles. Even with a high loading of 3.16 mg cm^−2^, the battery with a modified separator could still achieve a ~3.2 mAh cm^−2^ area capacity, which greatly increased its potential to meet the actual requirements.

## 2. Results and Discussion

As shown in Figure 1a, the synthesis of the NiCo_2_O_4_ sample went through three stages: template preparation, precursor adsorption and calcination (template removal). Through the classical hydrothermal reaction, the carbon sphere template with a smooth surface and uniform particle size (Appendix A) was obtained [21]. In the subsequent precursor coating process, Ni and Co ions were firmly adsorbed by various functional groups on the carbon surface, and formed a tight coating layer (Appendix A). Finally, the calcination in the air not only removed the carbon substrate, but also enabled the precursor on the surface to obtain an abundant pore structure.

The phase and purity of the as-obtained samples were first determined by X-ray powder diffraction (XRD) in Figure 2a. All patterns matched those of spinel NiCo_2_O_4_ (JCPDS No. 20-0781). The diffraction peaks at about 18.9°, 31.1°, 36.7°, 38.4°, 44.6°, 55.4°, 59.1°, and 64.9° corresponded to the (111), (220), (311), (222), (400), (422), (511), and (440) planes, respectively. The absence of other peaks meant that the sample was pure. Meanwhile, combined with the thermogravimetric analysis (TGA) results in Appendix A, it can be inferred that the carbon template is basically removed after a long time of heat treatment in air. Figure 2b depicts the scanning electron microscopy (SEM) image of NiCo_2_O_4_. Obviously, the samples maintained a spherical shape similar to that of the precursors that were about 400 nm in diameter. From the transmission electron microscopy (TEM) in Figure 2c, it can be seen that the edelweiss shaped nanospheres were hollow structures. The walls of these spheres were made up of a large number of ultra-thin NiCo_2_O_4_ nanosheets. In addition, closer inspection revealed that the nanosheets had a rich pore structure, which would greatly improve the surface area of the materials. Appendix A shows the SEM and TEM elemental mapping results. Obviously, the Ni, Co, and O elements were continuously and uniformly distributed in the as-prepared hollow NiCo_2_O_4_ sample. This once again proved the purity and image of the NiCo_2_O_4_ sample. The high-resolution TEM (HRTEM) in Figure 2d had the lattice fringes of 0.47 nm and 0.28 nm, which corresponded to the (111) plane and (220) plane of the spinel NiCo_2_O_4_, respectively.

The Brunauer–Emmett–Teller (BET) surface area of NiCo_2_O_4_ was determined by the N_2_ adsorption–desorption isotherm, as shown in Figure 2e. A type-IV curve with a pronounced hysteresis loop could be observed, indicating the high porosity of the sample [22]. Surprisingly, the BET surface area could be calculated as 281 m^2^ g^−1^, which was larger than the NiCo_2_O_4_ nanostructures reported in most of the other literature [20,22,23]. This might be due to the abundance of nanopores on the nanosheets. The pore size distribution at about 3.4 nm (Figure 2f) indicated that the sample had good mesoporous properties. In general, an extremely high surface area and abundant pore structure would greatly improve the adsorption site of NiCo_2_O_4_ to LiPSs and the infiltration and transfer rate of electrolyte and charge [24,25].

Using the slurry coating method, the fluffy hollow NiCo_2_O_4_ nanospheres were uniformly coated on one side of the commercial PP separator. As shown in Figure 2g, the other side of the modified separator remained intact. As a qualified separator modification layer, the composite NiCo_2_O_4_@PP should have a good electrolyte affinity, that is, it should not hinder the diffusion of electrolyte [16,26]. For this purpose, the contact angle experiments of the Li–S electrolyte on different separators were tested (Figure 2j,i). Obviously, the contact angle between the electrolyte and NiCo_2_O_4_@PP sample was ~6.4°. However, for the pure PP separator, the contact angle was as high as ~14.4°, indicating that the prepared NiCo_2_O_4_ not only did not hinder the diffusion of the electrolyte, but provided a better infiltrative property, which would be conducive to the kinetic acceleration of the electrochemical system [27].

In order to explore the anchoring effect and mechanism of NiCo_2_O_4_ on soluble LiPSs, a series of theoretical calculations and experimental verification were carried out. As shown in Figure 3a, the interaction between the soluble LiPSs (Li_2_S_4_, Li_2_S_6_, and Li_2_S_8_) and NiCo_2_O_4_ substrate was obvious, and the corresponding binding energies were 2.88 eV, 3.28 eV, and 3.10 eV, respectively. Importantly, the geometric configuration of these polysulfide molecules on the substrate was stable, indicating that NiCo_2_O_4_ was relatively stable for the electrochemical systems and chemical interactions [28,29]. In contrast, the binding energies of LiPSs on the carbon surface were less than 0.6 eV (Figure 3b), only 0.52 eV (Li_2_S_4_), 0.45 eV (Li_2_S_6_), and 0.42 eV (Li_2_S_8_), respectively. From the binding geometries in Appendix A, it can be seen that the LiPSs on the carbon surface were relaxed without obvious bonding cooperation. According to the reports of Zhang and Nazar et al., the weak interaction might be due to the van der Waals forces between molecules [14,15]. The different affinities of NiCo_2_O_4_ and carbon for LiPSs were reflected in the visual adsorption experiment (Figure 3c). After adding NiCo_2_O_4_ to the solution, it could be clearly observed that the color of the Li_2_S_4_ solution changed from brown to colorless. For activated carbon of the same mass, the color hardly changed at all. The strong adsorption of LiPSs by the former would contribute to the inhibition of the shuttle effect in the NiCo_2_O_4_@PP system.

Figure 3d,e and Appendix A reveal the X-ray photoelectron spectroscopy (XPS) results of NiCo_2_O_4_ before and after interacting with Li_2_S_4_. The XPS survey spectra (from 0 to 1300 eV) in Appendix A show that only the sulfur element changed before and after LiPS adsorption, indicating that no other impurities were involved in the reaction. In detail, for the pure NiCo_2_O_4_ sample, the Co 2p_3/2_ spectrum exhibited a broad peak consisting of two spin-orbit doublets, which corresponded to Co^2+^ and Co^3+^ [30,31]. The Co 2p_3/2_ spectrum were also fitted in a similar way. Two components at about 855.5 eV and 835.8 eV corresponded to Ni^3+^ and Ni^2+^, respectively. The wide peaks between 360 and 362 eV in binding energy were the corresponding satellite peaks [23]. After the adsorption of Li_2_S_4_, all the characteristic peaks shifted toward a lower binding energy, meaning that the Ni and Co cations interacted strongly with LiPSs, which was consistent with the results of previous studies [32,33,34,35]. In addition, the reduction in the Ni^3+^ and Co^3+^ contents suggested a partial reduction in the oxidation states of Ni and Co, which might be attributed to the charge transfer from S_4_^2−^ [34].

Subsequently, using the Li metal as the anode, the effect of the NiCo_2_O_4_@PP composite separator on the dynamics of a Li–S battery system was tested. Cyclic voltammetry (CV) curves of the carbon nanotube base sulfur cathodes with primitive PP or a NiCo_2_O_4_@PP separator conducted in a potential window of 1.7–2.6 V are presented in Figure 4a. Both cells exhibited classic LiPS electrochemical reaction characteristics. The absence of other peaks indicated that the electrode process was derived from the reaction with lithium and sulfur. Two cathode peaks could be attributed to the formation of soluble long-chain LiPSs and solid short-chain Li_2_S_2_/Li_2_S by sulfur reduction, respectively. The wide anode peak at about 2.3–2.4 V represented the oxidation of Li_2_S_2_/Li_2_S to S_8_ [36]. The voltage polarization of the cell using the NiCo_2_O_4_@PP composite separator was low, and the potential difference between the cathode peak and the anode peak was only 0.299 V. However, the cathode system with the initial PP separator was as high as 0.337 V, indicating that NiCo_2_O_4_ promoted the redox kinetics of the LiPS electrochemical system [37]. Figure 4b,c shows the Tafel plots and fitted slope values for the oxidation and the second reduction peaks in the CV curves in Figure 4a, respectively. Slopes as low as 85 and 107 mV dec^−1^ implied that the NiCo_2_O_4_ system had a better LiPSs conversion rate, which might be due to catalytic behavior at active sites rich in Ni and Co [38,39]. In this case, the accumulation of LiPSs would be avoided [34,40].

Figure 4d reveals the charge–discharge plots of different separator systems at 0.2 C. A higher reversible capacity of 1386 mAh g^−1^ was reached by the NiCo_2_O_4_@PP electrode. The voltage difference (peak spacing, Δ*E*) of NiCo_2_O_4_@PP was 0.165 V, lower than that of the PP electrode (0.196 V), which was consistent with the results of the CV. This might be due to the large number of active sites NiCo_2_O_4_ provides for the catalytic conversion of polysulfides [33,35]. From the CV and Tafel slope results, it can be seen that the NiCo_2_O_4_@PP had a relatively lower electrode process obstacle, faster charge/electron transfer rate, and better catalytic performance and redox kinetics. Therefore, in the charge–discharge view, it exhibited a low redox potential. Meanwhile, in the charging process, the initial overpotential of the pure PP system was slightly higher than that of NiCo_2_O_4_@PP (Figure 4e), indicating that the energy barrier of Li_2_S_2_/Li_2_S decomposition in the later system was low [41]. Electrochemical impedance spectroscopy (EIS) in Figure 4f (10 mHz–100 kHz) showed a small NiCo_2_O_4_@PP semicircle, which suggests that its charge-transfer resistance (*R_ct_*) was lower than the PP electrode. Here, the semicircle in the high frequency region corresponded to the charge-transfer step, while the straight line in the low frequency region corresponded to ion diffusion. After fitting (inset of Figure 4f) and calculation, the *R_ct_* of the NiCo_2_O_4_@PP electrode was ~14.6 ohm, while the original PP electrode was ~29.5 ohm. In addition, the surface resistance (*R_suf_*) of the former was only ~7.1 ohm, while the later was ~12.1 ohm. All of these electrochemical properties indicate that the NiCo_2_O_4_ modified layer had an excellent promotion effect on the reaction kinetics of the cell.

In terms of battery performance, the above dynamic improvement was manifested in the rapid charging–discharging ability. Figure 4g shows the comparison of the rate. At 0.2 C, 0.5 C, 1 C, 2 C, and 4 C, the discharge capacities of the NiCo_2_O_4_@PP-based cell were 1386, 982, 821, 656, and 505 mAh g^−1^, respectively, while the pure PP electrode was only 1246, 876, 731, 583, and 102 mAh g^−1^. The retention of the electrode capacity at high rate (such as 4 C) indicated that the NiCo_2_O_4_ coating could support the rapid charging–discharging of the cell. Meanwhile, when the current density returned to 0.2 C, the former still reached 1122 mAh g^−1^, higher than that of the PP electrode (1011 mAh g^−1^). Figure 4h displays the corresponding charging–discharging profiles at various rates. It could be seen that it retained the platform characteristics of LiPS reactions, especially at the high magnification of 4 C. Apart from that, although the polarization degree of the battery increased with the increase in the current density, the NiCo_2_O_4_@PP electrode always maintained a low voltage polarization, suggesting that the as-obtained NiCo_2_O_4_ effectively improved the cathode dynamics (Figure 4i), which was consistent with the data of CV and EIS. For the pure PP separator, the large polarization meant that the execution of the LiPS reaction at this rate was difficult [42,43].

Finally, to verify the improvement in the stability of the as-assembled Li−S cell by the NiCo_2_O_4_ modified separator, that is, the inhibition effect of the shuttle effect, various charge–discharge cycle experiments were conducted in Figure 5. At 0.5 C, the NiCo_2_O_4_@PP-based Li–S cell retained a considerable capacity of 702 mAh g^−1^ over 100 cycles (Figure 5a). The Coulombic efficiency gradually increased to ~100% in the first five cycles and remained at ~99.3% at the 100th. In the initial phase of the cyclic stability experiment, there was a gradual increase in capacity. This might be due to the low capacity caused by insufficient activation of the active substance at the beginning of cell operation. After a period of activation, the electrolyte was fully immersed into the cathode, and the sulfur was transformed from the initial status to a more electrochemically stable state. As a result, the electrodes exhibited high capacity [44,45]. Therefore, the cell capacity gradually reached its maximum. For the PP system, the capacity was only 501 mAh g^−1^ after 100 cycles. Impressively, when the current density was increased to 4 C, the former could run stably for more than 300 turns. As shown in Figure 5c, it could get up to a whopping 505 mAh g^−1^ capacity after activation. After 300 cycles, the capacity attenuation of each cycle of the Li–S cell with the NiCo_2_O_4_@PP separator was only ~0.064%. The Coulombic efficiency was stable around 99.4%. More importantly, the charge–discharge curve in Appendix A during the above cycling remained a good redox platform, showing excellent cyclic stability.

In addition to the rate and stability, the high sulfur loading is another important criterion of the Li–S cell [46,47]. Figure 5c exhibits the cycling performance of the NiCo_2_O_4_@PP-based cathode at a high sulfur loading of 3.16 mg cm^−2^. The maximum area capacity of about 3.2 mAh cm^−2^ (1012 mAh g^−1^, 0.2 C) was obtained and maintained at about 2.2 mAh cm^−2^ in subsequent cycles. The excellent electrochemical performance was not only due to the adsorption and blocking of LiPSs by the NiCo_2_O_4_ hollow nanospheres, but also due to the enhancement of redox reaction kinetics in the battery system by the abundant active catalytic sites provided by the huge specific surface area. Subsequently, the cell was disassembled after the cyclic stability test, and the composite separator was characterized by SEM. As shown in Appendix A, after long-term charge and discharge cycling, disassembly and cleaning, the surface of the modified separator was still covered with a large amount of NiCo_2_O_4_ matrix composite material, indicating that the obtained composite membrane showed excellent mechanical stability.

## 3. Materials and Methods

### 3.1. Preparation of Hollow NiCo_2_O_4_ and Modified Separator

For the carbon nanosphere templates, 10 g of glucose was dissolved in 40 mL deionized water to form a clarified solution. After 6 h of hydrothermal reaction at 160 °C, the reactor was cooled naturally. The brown precipitates were washed with water and ethanol three times, centrifuged, and dried in vacuum.

Subsequently, in the synthesis of hollow NiCo_2_O_4_ nanospheres, about 50 mg of the carbon nanospheres prepared above were dispersed ultrasonically in a mixture of 40 mL deionized water and 20 mL ethanol. A total of 0.29 g of Ni(NO_3_)_2_•6H_2_O, 0.582 g of Co(NO_3_)_2_•9H_2_O and 6 mmol of urea (0.36 g) were added and stirred for 1 hour. After 6 h of reaction in the water bath at 80 °C, it was cooled to room temperature. The precursors were separated by centrifugation, washed with deionized water and ethanol, respectively, and dried at 80 °C. Finally, the dried powder was calcined in air at 350 °C for 2 h with 0.5 °C min^−1^ to obtain the hollow NiCo_2_O_4_ product.

For a NiCo_2_O_4_@PP separator. The slurry consisting of NiCo_2_O_4_, polyvinylidene fluoride (PVDF) binder, and a carbon nanotube conductive agent with a weight ratio of 7:2:1 was coated on one side of a commercial Celgard-2400 PP separator. Then, the as-obtained modified separator was dried under vacuum.

### 3.2. Materials Characterization

The SEM and TEM (including HRTEM) images were carried out with a Hitachi Regulus 8100 and FEI Tecnai G^2^ 20, respectively. The XRD pattern was observed on a Rigaku D/Max III diffractometer with Cu Kα radiation (λ = 1.5418 Å). The specific surface area and pore size distribution were measured with a Kubo-X1000 analyzer with the BET calculation. The contact angle test was completed by JY-82B Kruss DSA. The XPS results were obtained using a Thermo Scientific K-Alpha tester. TGA was performed using a NETZSCH ASAP2020 thermal analyzer under the protection of air gas.

### 3.3. Visualization Li_2_S_4_ Adsorption

Briefly, sulfur and Li_2_S with a molar ratio of 3:1 was added to a mixed solution of 1,3-dioxolane (DOL) and 1,2-dimethoxymethane (DME) at 50 °C with stirring. The Li_2_S_4_ solution concentration was controlled at about 0.5 M. The same mass of carbon nanotubes and hollow NiCo_2_O_4_ nanospheres were added to the above solution, respectively.

### 3.4. Electrochemical Measurements

The sulfur composite cathode was prepared by melting sublimated sulfur power (70 wt%) and carbon nanotubes (30 wt%) at 155 °C in an airtight reactor. The resulting sulfur-based composite was then mixed with acetylene black and PVDF in a N-methyl-pyrrolidone (NMP) solvent at a mass ratio of 7:2:1 and evenly coated on clean Al foil. For the general tests, the sulfur loading was controlled at about 1 mg cm^−2^. For the high loading tests, the sulfur loading was 3.16 mg cm^−2^. Subsequently, Li–S cells (CR2032) were assembled in a glove box filled with Ar. Lithium foil and pure PP or NiCo_2_O_4_@PP were used as the anode and separator, respectively. The DOL/DME (*v*/*v*, 1:1) solvent with 1 M lithium (trifluoromethylsulfonyl) imide (LiTFSI) and 0.1 M LiNO_3_ was used as the electrolyte. The removal of the battery all took place in the glove box and the separator sample was cleaned with DME.

All constant-current charge–discharge tests were performed on a LAND system with a voltage range of 1.7–2.6 V. The CV and EIS were tested on an electrochemical workstation (CHI 660E).

### 3.5. Theoretical Calculation

All theoretical calculations were conducted using the DMol^3^ system. Density functional theory was used to calculate the LiPS adsorption on the NiCo_2_O_4_ matrix. The Perdew–Burke–Ernzerhof form of the exchange-correlation effects was employed. Monkhorst–Pack mesh k-points were used to sample the Brillouin zone with a cut-off energy of 300 eV. The binding energy (E_b_) can be calculated as E_b_ = E_sub_ + E_ps_ − E_sub+ps_, where E_sub_, E_ps_, and E_sub+ps_ represent the ground-state energies of the substrate, polysulfide, and substrate-polysulfide, respectively.

## 4. Conclusions

In summary, the edelweiss shaped hollow nanospheres assembled by ultra-thin NiCo_2_O_4_ nanosheets were prepared by a simple template method. The specific surface area of the product was up to ~281 m^2^ g^−1^ due to the hollow inner cavity and the surface nanosheet structure. This provided a wealth of active sites for the anchoring and catalytic conversion of LiPSs. As the modification layer of the separator, the prepared NiCo_2_O_4_ nanostructures not only physically hindered the diffusion of LiPSs and limited their electrochemical reaction, but also chemically interacted with LiPSs and the electrolyte and promoted the reaction kinetics of the electrode. As a result, the Li–S cell with the NiCo_2_O_4_@PP separator exhibited an amazing capacity of 1386 mAh g^−1^ at 0.2 C and a high capacity of 505 mAh g^−1^ at 4 C. Even after 300 cycles, it still had a commendable capacity retention of ~80.6%. Generally, we demonstrated a multifunctional NiCo_2_O_4_@PP composite separator that enabled a Li–S battery with high capacity, good rate, and stable cycling life.

## Figures and Tables

**Figure 1 molecules-28-01539-f001:**
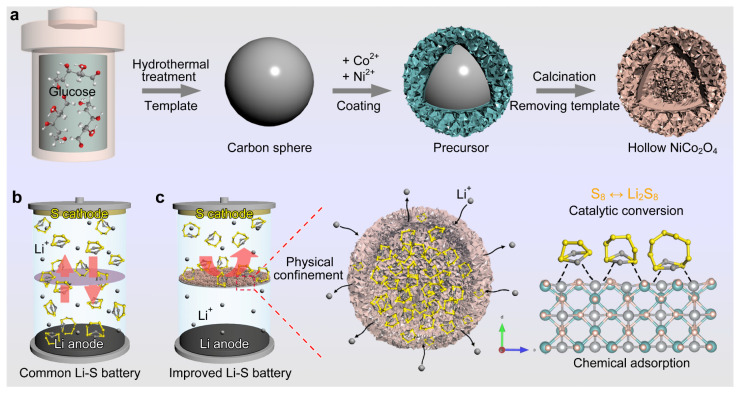
Schematic diagrams of (**a**) the synthesis process of NiCo_2_O_4_ hollow nanospheres, (**b**) a Li–S battery with a common PP separator, and (**c**) an improved Li–S system using a NiCo_2_O_4_@PP separator for minimizing the issues of LiPSs.

**Figure 2 molecules-28-01539-f002:**
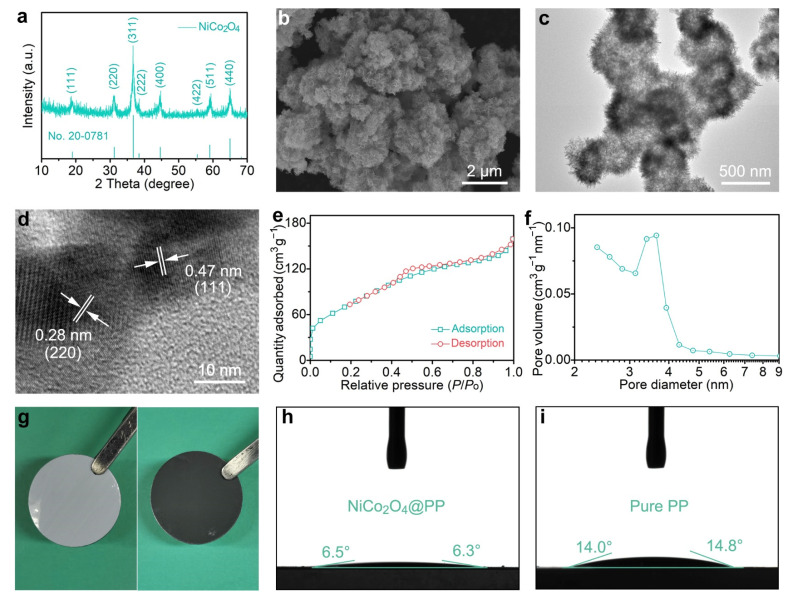
(**a**) XRD, (**b**) SEM, (**c**) TEM, and (**d**) HRTEM characteristics of the as-obtained NiCo_2_O_4_ sample. (**e**,**f**) N_2_ adsorption–desorption isotherm and pore size distribution. (**g**) Photographs of the NiCo_2_O_4_@PP separator. (**h**,**i**) Contact angle tests of the modified and pure separators.

**Figure 3 molecules-28-01539-f003:**
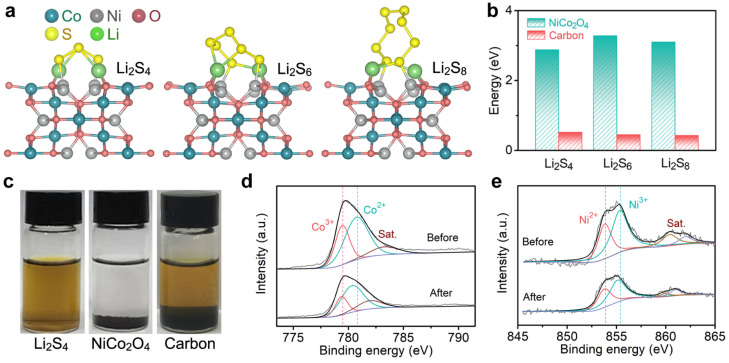
(**a**) Mechanism diagram of interaction between NiCo_2_O_4_ and LiPSs. (**b**) The corresponding adsorption energies. (**c**) Visual adsorption of Li_2_S_4_. (**d**) Co 2p and (**e**) Ni 2p XPS results of NiCo_2_O_4_ before and after Li_2_S_4_ adsorption.

**Figure 4 molecules-28-01539-f004:**
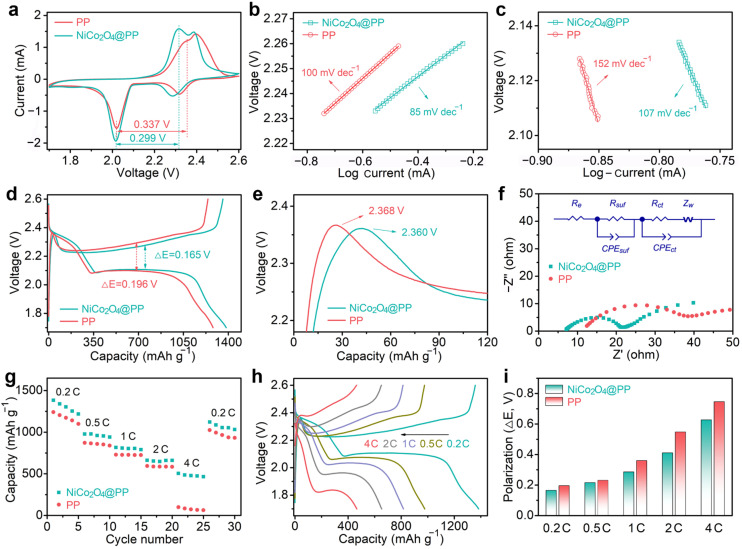
(**a**) The CV curves with pure PP and modified separators at 0.1 mV s^−1^. (**b**,**c**) Corresponding Tafel results and fitted slope values. (**d**,**e**) Comparison of the charge–discharge plots at 0.2 C, and the corresponding initial process of the charge details. (**f**) EIS spectra. The inset is the corresponding equivalent circuit. *R_e_* is the electrolyte resistance. *R_suf_* is the surface resistance. *R_ct_* is the charge transfer resistance. *Z_w_* is the Warburg impedance. The constant phase element (CPE) was used to compensate for the non-ideal behavior of the electrode. (**g**,**h**) Rate performance and charge–discharge curves at different currents. (**i**) Comparison of the potential polarizations of different separators.

**Figure 5 molecules-28-01539-f005:**
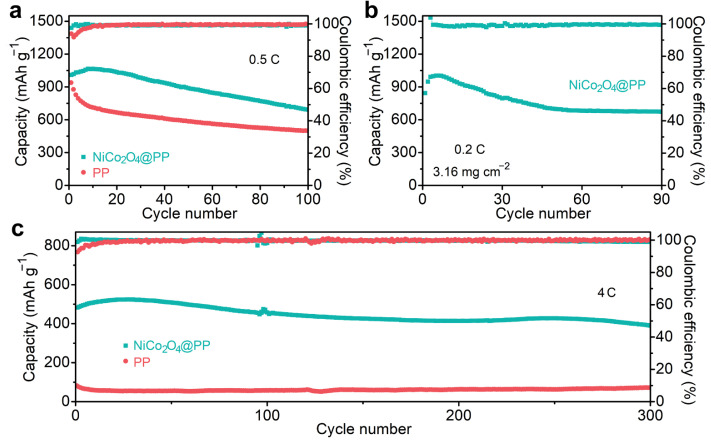
(**a**) Cyclic properties of the sulfur cathodes with different separators at 0.5 C. (**b**) High loading performance at 0.2 C. (**c**) High rate and long cycle comparison.

## Data Availability

Not applicable.

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
