# Peer review of "Multiple Effects of High Surface Area Hollow Nanospheres Assembled by Nickel Cobaltate Nanosheets on Soluble Lithium Polysulfides"

_molecules, 2023, doi:10.3390/molecules28041539_

Round 1

Reviewer 1 Report

Dear editor,

I read with interest this manuscript reporting on a novel separator for Li-S batteries. This paper in my view represents an important contribution to the field. The claims are well supported by the complementary characterization results. As such, I recommend publication of the paper after minor revisions.

1. Why the modification leads to a lower redox potential of sulfur in Fig 4.?

2. Is it possible the NiCo2O4 oxides the sulfur species so that the color disappears in Fig. 3c?

Author Response

Reply to Reviewer #1

Reviewer #1: I read with interest this manuscript reporting on a novel separator for Li-S batteries. This paper in my view represents an important contribution to the field. The claims are well supported by the complementary characterization results. As such, I recommend publication of the paper after minor revisions.

Comment 1:

Why the modification leads to a lower redox potential of sulfur in Fig. 4?

Reply:

Thanks for very much for your kind support on our manuscript.

The NiCo2O4@PP composite separator has a lower polarization potential, indicating that the NiCo2O4 coating effectively reduces the polarization of the cell. This may be due to the large number of active sites NiCo2O4 provides for the catalytic conversion of polysulfides (Adv. Energy Mater. 2019, 9, 1803477; Adv. Energy Mater. 2019, 9, 1900228; Nano Lett. 2021, 21, 5285−5292). From the CV, Tafel slope and EIS impedance results, it can be seen that the NiCo2O4@PP had relatively lower electrode process obstacle, faster charge/electron transfer rate, better catalytic performance and redox kinetics. Therefore, in the charge-discharge view, it has a low redox potential.

Comment 2:

Is it possible the NiCo2O4 oxides the sulfur species so that the color disappears in Fig. 3c?

Reply:

Thank you for this good question.

We performed XPS characterization of S element in the adsorbed sample, as shown in Figure R1. After the adsorption of LiPSs, the surface of NiCo2O4 still has obvious terminal sulfur (ST1) and bridging sulfur (SB0) peaks, which belong to LiPSs (Adv. Energy Mater. 2016, 6, 1501636; Nat. Commun. 2014, 5, 4759; Nat. Commun. 2015, 6, 5682). This suggests that the LiPSs (coloured) is still present on NiCo2O4.

In addition, as reported in UV-VIS spectra in the previous literature (Adv. Energy Mater. 2019, 9, 1900228; Adv. Energy Mater. 2019, 9, 1803477; Nano Lett. 2021, 21, 5285−5292), S4 decreased significantly in the adsorbent solution.

According to these, it can be inferred that the color receding may not be due to the oxidation process, but from the adsorption process.

Figure R1. XPS spectrum of S 2p for NiCo2O4/Li2S4.

Reviewer 2 Report

     In this review, “Multiple Effects of High Surface Area Hollow Nanospheres Assembled by Nickel Cobaltate Nanosheets on Soluble Lithium Polysulfides”, the authors utilized porous hollow spheres of NiCo2O4 to modify the Celgard separator to address the issue of dissolved polysulfide shuttles in the lithium-ion batteries.  However, the present manuscript should be revised before its publication in “Molecules” to further improve the quality.

1.      The authors claim the complete removal of carbon sphere upon calcination in air. To prove it, the TGA of pristine carbon sphere and the composite should be provided in the revised manuscript.

2.      The XPS survey spectrum should be included in the revised supporting information.

3.      SEM & TEM elemental mapping should be provided to know the elemental distribution on the hollow spheres.

4.      In Figure 2g, the uniform coating of hollow spheres over the separator is demonstrated. However, its mechanical stability is an important parameter for the device application. So, the condition of coating should be investigated by dis-assembling the cell after the cyclic stability test. The respective data should be included in the revised manuscript.

5.      In Figure 4f (EIS data), the limits of X- and Y-axes should be the same (both axes should be plotted to 50 Ohm).

6.      The impedance data should be fitted and the respective circuit model should be provided as an inset.

7.      In the cyclic stability experiment, the reason for the initial increment of the specific capacity for the modified separator is not properly explained in the main text. A detailed discussion should be added in the revised manuscript.

8.      Some of the important references should be included in the revised manuscript.

(i). Energy Storage Mater. 2019, 23, 707-735 (10.1016/j.ensm.2019.02.022)

(ii). 2D Mater. 2022, 9, 034002 (10.1088/2053-1583/ac7056)

(iii). Batteries & Supercaps 2022, 5, e202200097 (10.1002/batt.202200097)

Author Response

Reply to Reviewer #2

Reviewer #2: In this review, “Multiple Effects of High Surface Area Hollow Nanospheres Assembled by Nickel Cobaltate Nanosheets on Soluble Lithium Polysulfides”, the authors utilized porous hollow spheres of NiCo2O4 to modify the Celgard separator to address the issue of dissolved polysulfide shuttles in the lithium-ion batteries. However, the present manuscript should be revised before its publication in “Molecules” to further improve the quality.

Comment 1:

The authors claim the complete removal of carbon sphere upon calcination in air. To prove it, the TGA of pristine carbon sphere and the composite should be provided in the revised manuscript.

Reply:

Thanks for very much for your kind support on our manuscript.

Figure S2 shows the TGA results of pure carbon sphere and precursor composite. It can see that at 350 °C, the carbon is already breaking down. Meanwhile, combined with the spectra of carbon-free peaks in XRD, it can be inferred that after a long time of heat treatment, carbon will be removed.

Figure S2. TGA results of (a) pure carbon sphere, and (b) precursor composite.

Comment 2:

The XPS survey spectrum should be included in the revised supporting information.

Reply:

Thanks for your suggestion.

In revised manuscript, the XPS survey spectra were added in Figure S5. It showed that only sulfur element changed before and after LiPSs adsorption, indicating that no other impurities were involved in the reaction.

Figure S5. XPS survey spectra of NiCo2O4 before and after Li2S4 adsorption.

Comment 3:

SEM & TEM elemental mapping should be provided to know the elemental distribution on the hollow spheres.

Reply:

Thanks for your valuable comment.

In revised supporting information, Figure S3 shows the SEM and TEM elemental mapping. Obviously, the Ni, Co and O elements are continuously and uniformly distributed in the as-prepared hollow sample. The corresponding description has been added to the text.

Figure S3. Elemental mapping of NiCo2O4: (a) SEM, (b) TEM.

Comment 4:

In Figure 2g, the uniform coating of hollow spheres over the separator is demonstrated. However, its mechanical stability is an important parameter for the device application. So, the condition of coating should be investigated by dis-assembling the cell after the cyclic stability test. The respective data should be included in the revised manuscript.

Reply:

Thanks for your suggestion. 

The battery was disassembled after the cyclic stability test, and the composite separator was characterized by SEM. As shown in Figure S7, after long-term charge-discharge cycling, disassembly and cleaning, the surface of modified separator was still covered with a large amount of NiCo2O4 matrix composite material, indicating that the obtained composite membrane showed excellent mechanical stability.

Figure S7. The surface SEM of composite separator after cyclic test.

Comment 5:

In Figure 4f (EIS data), the limits of X- and Y-axes should be the same (both axes should be plotted to 50 Ohm).

Reply:

Thanks for your valuable suggestion. We restricted the X- and Y- axes in Figure 4f to the same range.

Figure 4f. (f) EIS spectra of different separators.

Comment 6:

The impedance data should be fitted and the respective circuit model should be provided as an inset.

Reply:

Thanks for your suggestion.

In the revised manuscript, the impedance data was fitted and the corresponding equivalent circuit model was provided as an inset in Figure 4f.

Here, Re was the electrolyte resistance. Rsuf referred to the surface resistance. Rct represented the charge transfer resistance. Zw meant the Warburg impedance. Constant phase element (CPE) was used to compensate for non-ideal behavior of electrode (J. Energy Chem. 2022, 68, 762–770). After fitting, Rs and Rct of the composite NiCo2O4@PP separator were 7.1 ohm and 14.6 ohm, respectively, lower than those of the pristine PP based electrode (12.1 ohm and 29.5 ohm).

Figure 4f. (f) EIS spectra of different separators. The inset is the corresponding equivalent circuit.

Comment 7:

In the cyclic stability experiment, the reason for the initial increment of the specific capacity for the modified separator is not properly explained in the main text. A detailed discussion should be added in the revised manuscript.

Reply:

Thanks for your suggestion.

In the initial phase of the cyclic stability experiment, there was a gradual increase in capacity. This might be due to low capacity caused by insufficient activation of the active substance at the beginning of cell operation. After a period of activation, the electrolyte was fully immersed into the cathode, and the sulfur was transformed from initial status to a more electrochemically stable state. As a result, the electrodes exhibit high capacity (Adv. Mater. 2016, 28, 6926–6931; Electrochim. Acta 2017, 250, 159–166). Therefore, the cell capacity gradually reached its maximum.

Comment 8:

Some of the important references should be included in the revised manuscript.

(i). Energy Storage Mater. 2019, 23, 707-735 (10.1016/j.ensm.2019.02.022)

(ii). 2D Mater. 2022, 9, 034002 (10.1088/2053-1583/ac7056)

(iii). Batteries & Supercaps 2022, 5, e202200097 (10.1002/batt.202200097)

Reply:

Thanks for your suggestion.

We have added these important references (Ref. [4]. [5]. [6]) to the revised manuscript.

Round 2

Reviewer 2 Report

The revised manuscript could be accepted for publication in present form.